

# Catcalls: exotic cats discriminate the voices of familiar caregivers

Taylor Crews[1], Jennifer Vonk[1] and Molly McGuire[2]

[1] Department of Psychology, Oakland University, Rochester, MI, United States of America
[2] Animal Wellbeing, Zoo Miami, Miami, FL, United States of America

## ABSTRACT

**Background**. The ability to differentiate familiar from unfamiliar humans has been considered a product of domestication or early experience. Few studies have focused on voice recognition in *Felidae* despite the fact that this family presents the rare opportunity to compare domesticated species to their wild counterparts and to examine the role of human rearing.

**Methods**. We tested whether non-domesticated *Felidae* species recognized familiar human voices by exposing them to audio playbacks of familiar and unfamiliar humans. In a pilot study, we presented seven cats of five species with playbacks of voices that varied in familiarity and use of the cats' names. In the main study, we presented 24 cats of 10 species with unfamiliar and then familiar voice playbacks using a habituation-dishabituation paradigm. We anticipated that human rearing and use of the cats' names would result in greater attention to the voices, as measured by the latency, intensity, and duration of responses regardless of subject sex and subfamily.

**Results**. Cats responded more quickly and with greater intensity (*e.g.*, full versus partial head turn, both ears moved versus one ear twitching) to the most familiar voice in both studies. They also responded for longer durations to the familiar voice compared to the unfamiliar voices in the main study. Use of the cats' name and rearing history did not significantly impact responding. These findings suggest that close human contact rather than domestication is associated with the ability to discriminate between human voices and that less social species may have socio-cognitive abilities akin to those of more gregarious species. With cats of all species being commonly housed in human care, it is important to know that they differentiate familiar from unfamiliar human voices.

# INTRODUCTION

The ability to recognize familiar and unfamiliar individuals plays a pivotal role in the social lives of animals, and this is often accomplished through voice recognition. For example, mother cats (*Felis catus*) recognize the cries of their kittens (*Szenczi et al., 2016*), lions (*Panthera leo)* identify members within and outside of their social groups (*Gilfillan et al., 2016*), spotted hyenas (*Crocuta crocuta*) respond more strongly to the whoops of cubs to whom they are related (*Holekamp et al., 1999*), and female great tit birds (*Parus major*) discriminate their mate's call from neighbors' vocalizations (*Blumenrath,*

Corresponding author
Jennifer Vonk, vonk@oakland.edu

*Dabelsteen & Pedersen, 2007*). Beyond recognizing individual and familiar conspecifics, nonhuman primates (*Candiotti, Zuberbühler & Lemasson, 2013*) and birds (*Dhondt & Lambrechts, 1992*) can identify individuals of neighboring heterospecifics. With humans becoming a more commonly encountered heterospecific, animals may develop the ability to discriminate human voices, and this may depend upon the nature of exposure to humans. We tested whether members of various species of non-domesticated cats with different rearing histories differentiated between familiar and unfamiliar human voices.

Recognition of human vocal cues has been investigated in many domesticated species, such as dogs (*Canis lupus familiaris)* (*Adachi, Kuwahata & Fujita, 2007*), horses (*Equus caballus)* (*d'Ingeo et al., 2019*), pigs *(Sus scrofa domesticus)* (*Bensoussan et al., 2019*), and cats *(Felis catus)* (*Saito & Shinozuka, 2013*). The few studies that have examined non-domesticated species focused on animals that are naturally gregarious, such as gorillas *(Gorilla gorilla*, *Salmi, Jones & Carrigan, 2022*) and elephants (*Loxodonta africana*, *McComb et al., 2014*). In a review of human vocal discrimination in nonhumans (*Kriengwatana, Escudero & Ten Cate, 2015*), the only relatively asocial species represented was the domestic cat (*Saito & Shinozuka, 2013*). There is little investigation of vocal recognition in exotic cats despite their prevalence in human care. The current study extends this research to 25 individuals of 10 non-domesticated felid species housed in human care. The cat family, or *Felidae*, are of interest given their relatively asocial natural history coupled with their close association with humans in modern society. They are a highly diverse group consisting of 38 species (*IUCN Red List, 2021*). The subfamily, Pantherinae, consists of seven species: lions (*Panthera leo*), tigers (*Panthera tigris*), leopards (*Panthera pardus*), jaguars (*Panthera onca*), snow leopards *(Panthera uncia),* and two species of clouded leopard (*Neofelis nebulosa* and *Neofelis diardi)*. The subfamily, Felinae, which is responsible for the evolutionary line that produced the modern domestic cat, includes the remaining 31 species (*Castello, Sliwa & Kitchener, 2020*). Members of both Pantherinae and Felinae are commonly found in human care, housed in zoos, sanctuaries, nature preserves, and personal collections in great numbers. In the current studies, we included members of four Pantherinae species (clouded leopard, *Neofelis nebulosa*; snow leopard, lion, and tiger) and six Felinae species (cheetah, *Acinonyx jubatus;* cougar, *Puma concolor* cory; fishing cat, *Prionailurus viverrinus*; Canadian lynx, *Lynx canadensis;* sand cat, *Felis margarita* and serval, *Leptailurus serval*) allowing us to address the potential breadth of the ability to recognize familiar heterospecific voices across non-domesticated felid species.

Recognition of individuals along with other socio-cognitive abilities, such as the ability to follow gaze and point cues, have been attributed to the process of domestication (*Hare et al., 2002*; *Topál et al., 2005*). However, megachiropteran bats (*Pteropus*) were able to follow point signals to locate hidden food items only when socialized with humans from an early age (*Hall et al., 2011*), suggesting that socialization with humans might be as important as domestication (if not more so) in facilitating an understanding of human communicative behaviors. *Saito & Shinozuka (2013)* demonstrated that domestic cats respond differently to their owner's voice compared to the voices of unfamiliar humans. If wild cats share with domestic cats the ability to differentiate human voices, this would suggest that this ability is not dependent on domestication or human rearing (henceforth, hand-reared).

A single study conducted with wild cats (*Leroux et al., 2018*) found that a group of eight hand-reared male cheetahs (*Acinonyx jubatus*) discriminated between the voices of familiar and unfamiliar humans, as indicated by greater visual attention and more rapid response to familiar human voices compared to unfamiliar human voices. If social ecology is critical for the development of individual vocal recognition, lions may show stronger discrimination of familiar voices compared to other species, as they are the only wild cat known to live in large social groups (*Sunquist & Sunquist, 2002*). Here, we examine our results with lions both included and excluded from analyses to examine the impact of group-living on heterospecific recognition.

Because the single finding of human voice discrimination in non-domestic cats comes from a group of hand-reared cheetahs, it is important to determine whether the finding is dependent upon human rearing. Hand-reared cats may exhibit important differences from mother-reared cats. For example, *Mellen (1992)* found differences in responses to familiar and unfamiliar humans such as heightened aggression and fear, as well as less interest in social interaction to unfamiliar (relative to familiar) humans in human-reared compared with mother-reared kittens. Mellen's results have informed husbandry and rearing techniques in zoos, encouraging mother-rearing whenever possible. Whereas the preference for most species and facilities is that offspring are mother-reared (*Hampson & Schwitzer, 2016*), there are some circumstances that require human intervention (*e.g.*, maternal neglect, wild orphans, infanticide, or large litters that are physically taxing on the mother, *Hines, n.d.*). Thus, we were able to examine the role of hand-rearing compared to mother-rearing in response to human voices.

A secondary question was whether cats might respond to the sound of their names without regard for the identity of the caller. *Saito et al. (2019)* found that domestic cats responded more to their names than to other words but whether exotic cats are more responsive to commands or greetings when their names are spoken is unknown. The animals housed in managed care frequently work closely with their human caretakers, as a necessary aspect of their daily husbandry. Animal caretakers commonly use verbal cues when working with the animals, especially during training and when calling the animals by their names. These animals also may interact to some degree with the public regularly, so that they hear many different voices and words, including their own names if those are known to the public. There is much debate in the captive animal industry as to whether names of animals should be posted publicly for visitors to know (Crews, T., 2023, personal observation). The giving of a name to an animal is a form of anthropomorphism, in which humans might unconsciously assign characteristics and be more sympathetic to a named animal than to an unnamed animal (*Chartrand, Fitzsimons & Fitzsimons, 2008*). The constant repetition of a behavioral cue, such as the calling of a name, without a subsequent reinforcer may desensitize the animal to that cue, which could lead to complications in training and frustration for the animal (*Miltenberger, 2015*). If the cats are responsive to their names regardless of whether they are spoken to by a familiar or unfamiliar voice, it might be beneficial to prevent members of the public from knowing and calling the cats' names. At the same time, such a finding may demonstrate that cats will be more responsive to communication from their keepers if their names are used.

The purpose of this study was to examine recognition of familiar human voices in captively-managed exotic cats excluding the domestic cat, *F. catus*, as well as any hybrids of the domestic cat, such as savannah (*Felis catus × Leptailurus serval*) and bengal cats (*Felis catus × Prionailurus bengalensis)*. We conducted two studies using slightly different playback procedures. In a pilot study, we presented a small group of captive cats with six 3-trial sessions including playbacks of unfamiliar, less familiar, and more familiar caregiver voices either using their name or not. We were interested in whether any effects of familiarity were restricted to humans with which the cats had particularly close relationships. To include more cats from various facilities in the main study, we minimized the number of required sessions by shifting to the dishabituation paradigm used by *Saito & Shinozuka (2013)* and focused on only the most familiar caregiver's voice. This method allowed us to witness a stronger response to the familiar voice when directly contrasted with unfamiliar voices within the same sessions. Within session comparisons limit the influence of extraneous factors that vary across test sessions. In our main study, cats were presented with a series of audio playbacks of three different unfamiliar humans speaking the same phrase, then a playback of a familiar voice and, finally, a fourth unfamiliar voice. Typically, in this paradigm, subjects habituate to the sound of strangers' voices, but show a rebound effect in attention and responsiveness during the familiar voice playback, that then dissipates with a subsequent unfamiliar voice (*Saito & Shinozuka, 2013*). Such a pattern suggests recognition and discrimination of the familiar voice from among other voices. For our purposes, we did not consider it necessary that the cats habituated to the unfamiliar voices so long as they showed a pronounced response to the familiar voice that differed from that to the unfamiliar voices, because the cats were tested in public facilities where we could not completely control other sounds, including voices, during testing. We hypothesized that cats would show greater attention as measured by faster latencies to respond and greater intensity and duration of responses following the familiar voice relative to the unfamiliar voices. Behavioral responses such as head, ear, and body orientation, movement towards or away from the sound, and vocalizations were considered cumulatively rather than separately as was done by *Saito & Shinozuka (2013)*, due to the rarity of any single behavior. We compared responses to the five vocal cues in sessions in which the subjects' names were or were not spoken, and examined the predictive factors of sex, subfamily (*Pantherinae versus Felinae*), and rearing history (hand-reared *vs.* mother-reared). We predicted an interaction between rearing history and familiarity in that cats that were hand-reared would respond faster, more intensely and for longer to voices they were familiar with compared to cats that had been mother-reared. Lastly, we predicted that cats would respond faster, more intensely and for longer when the cues included their names, compared to the cues with no name spoken, and that the use of a name would interact with familiarity to predict the intensity of responses, such that the cats would show a stronger response to their name only when spoken by a familiar speaker. Inclusion of sex and subfamily were for exploratory purposes as we did not have specific hypotheses regarding these factors.

## MATERIAL AND METHODS

### Ethics statement

The experiments reported here were reviewed and approved by Oakland University's IACUC (Protocol # 2021-1155).

### Subjects

Seven individuals of five species (tiger, *P. tigris*; cheetah, *A. jubatus*; serval, *L. serval*; puma, *P. concolor*; lynx, *L. canadensis*) housed at Zoo Miami in Florida and the Creature Conservancy in Ann Arbor, MI participated in a pilot study. All but one of these cats, plus 18 additional cats representing ten species subsequently participated in the main study (approximately 3–6 months later). Information about each subject's subfamily (Pantherinae or Felinae), sex, and rearing history (hand-reared or mother-reared) are reported in Table 1. Rearing history was categorized as hand reared (reared exclusively by humans starting at no later than four months of age, which is the start of the weaning period for most large exotic cats; *Jhala & Sadhu, 2017*) or mother-reared (raised exclusively by their mother, or co-raised with the mother and humans, from birth to at least four months of age). We did not differentiate between subspecies; that is, Malayan tigers (*P. tigris tigris)* and Sumatran tigers (*P. tigris sumatrae)* were classified as tigers.

### Materials

Audio recordings of each voice were taken using a Zoom H1n Handy recorder (Zoom, San Jose, CA, USA) and played back for the subjects using an Ultimate Ears BOOM 3 Bluetooth speaker (Boom Technology, Dove Valley, CO, USA). Observations were recorded using a GoPro Hero 10 (GoPro, San Mateo, CA, USA). Data were coded and analyzed by naïve coders using freely available BORIS v.8.20. software (*Friard & Gamba, 2016*). All testing occurred at the cat's home facility, in their regular habitat. The researcher performed all playback sessions, as well as recorded all observations, from outside of the enclosure.

In the pilot study, each cat was exposed to three different voices speaking two different cue types. Thus, each human speaker provided six different recordings, three of them speaking the name absent (NA) cue, and three of them speaking the name present (NP) cue. A total of 18 recordings were collected for each cat to ensure that the cats never heard the exact same recording more than once to control for habituation. The speakers, matched for sex, were categorized as Most Familiar (MF), Less Familiar (LF), and Unfamiliar (UF). In the main study, each cat was exposed to five different voices (four UF and one MF voice) speaking the same two cue types (NA, NP). Thus, each speaker provided two different recordings, one for the NP condition, and one for the NA condition. The MF speaker was someone that the cat was very familiar with, such as the cat's primary keeper or trainer. If the facility did not assign primary trainers or keepers, then we used the voice of the person who had worked with the cat for the longest period of time. The LF speaker was someone that the cat had heard before, but had minimal structured interaction with. This person was either a member of staff that did not work with the cat, an intern, or a volunteer. The LF speaker could not have actively participated in training or care of the cat, which included but was not limited to, feeding, directly providing enrichment, or participating in

**Table 1** Subject subfamily, species common name, sex, rearing history, facility and study involvement.

| Subject | Subfamily | Species | Sex | Rearing style | Facility | Study participation | Sessions -Pilot | Sessions -Main |
|---|---|---|---|---|---|---|---|---|
| Diesel | Felinae | Cheetah | M | HR | Zoo Miami | 1 and 2 | 3 | 3 |
| Koda | Felinae | Cheetah | M | HR | Zoo Miami | 1 and 2 | 3 | 3 |
| Nia | Felinae | Cheetah | F | HR | Cincinnati Zoo | 2 | | 2 |
| Tommy | Felinae | Cheetah | M | HR | Cincinnati Zoo | 2 | | 2 |
| Jean | Pantherinae | Clouded Leopard | F | HR | Zoo Miami | 2 | | 2 |
| Harper | Felinae | Cougar | F | HR | Creature Conservancy | 1 and 2 | 6 | 2 |
| Tecumseh | Felinae | Cougar | M | HR | Cincinnati Zoo | 2 | | 4 |
| Joey | Felinae | Cougar | M | HR | Cincinnati Zoo | 2 | | 4 |
| Tallulah | Felinae | Fishing Cat | F | MR | Greensboro Science Center | 2 | | 3 |
| Mako | Felinae | Fishing Cat | M | MR | Greensboro Science Center | 2 | | 3 |
| Gordie | Felinae | Canadian lynx | M | HR | Creature Conservancy | 1 and 2 | 5 | 2 |
| Amirah | Pantherinae | Lion | F | MR | Zoo Miami | 2 | | 4 |
| Imani | Pantherinae | Lion | F | HR | Cincinnati Zoo | 2 | | 2 |
| John | Pantherinae | Lion | M | MR | Cincinnati Zoo | 2 | | 2 |
| Layla | Felinae | Sand Cat | F | MR | Greensboro Science Center | 2 | | 2 |
| Kira | Felinae | Serval | F | HR | Greensboro Science Center | 2 | | 3 |
| Major | Felinae | Serval | M | HR | Zoo Miami | 1 and 2 | 4 | 2 |
| Scout | Felinae | Serval | M | HR | Zoo Miami | 1 and 2 | 4 | 2 |
| Tut | Felinae | Serval | M | HR | Greensboro Science Center | 2 | | 3 |
| Nubo | Pantherinae | Snow Leopard | M | MR | Cincinnati Zoo | 2 | | 2 |
| Renji | Pantherinae | Snow Leopard | F | HR | Cincinnati Zoo | 2 | | 2 |
| Berani | Pantherinae | Tiger | M | MR | Zoo Miami | 1 | 3 | |
| Jin | Pantherinae | Tiger | F | MR | Cincinnati Zoo | 2 | | 3 |
| Rocky | Pantherinae | Tiger | M | MR | Greensboro Science Center | 2 | | 2 |
| Zero | Pantherinae | Tiger | M | HR | Cincinnati Zoo | 2 | | 3 |

**Notes.**
HR, human-reared; MR, mother-reared.

husbandry tasks such as veterinary care. The UF speakers were four different sex-matched people that the cat had never encountered or heard before. These recordings were provided by individuals that had not been to the facility the cat currently resided at or had resided at in the past.

The NA cue was a short phrase that the cat was familiar with, "Good morning, how are you doing today?". This phrase was selected from the results of a poll from exotic cat keepers on zookeeping Facebook pages, in which they submitted a phrase they speak regularly to the cats in their care but that was not associated with food rewards. Some variation of the used phrase was found to be the most common occurrence across facilities. The NP cue matched the NA cue, but the cat's name was said in the phrase, *e.g.*, "Good morning, Harper, how are you doing today?". In instances where the cat had multiple names, such as registered names, house names, or public names, the name that was used was the one that the animal care staff used most frequently. We deliberately used a phrase that we knew was familiar to all cats given that we could not be aware of the cats' exposure

to less common phrases, and because this might be the strongest test of whether the cats differentiated familiar voices from unfamiliar voices speaking familiar phrases.

All recordings were made in a quiet room with no additional voices in the background. The recordings were less than 5 s long. The speakers were controlled for sex, volume, and neutral tone of voice, matching the MF voice within 1 dB and 2 Hz. In the main study, Audacity v.3.4.2 audio editing software was used to create a single file for each session of playbacks with the order of the UF voices randomized for each subject to control for reactions to specific voices occurring in the same temporal position across sessions.

## Procedure

A wireless speaker was set up outside of the enclosure in the visitor area, and was no closer than three feet away from the primary containment barrier. The researcher and observation recording equipment were also outside of the enclosure in visitor space (Fig. 1). Prior to each trial, the cat must have been on exhibit for at least 15 min and must have been awake. The time allotment was to avoid potential distraction or inattention due to examining the space for food or enrichment items. Each subject was provided an acclimation period with the researcher and the equipment. Sessions did not start until the subject(s) had shown no interest or attention to the recording equipment or researcher for at least two minutes. All trials were performed in the absence of visitors, either before or after operating hours, during lulls in visitation, or on days the facility was not open to the public. Eight cats were housed in pairs; lions, snow leopards and cougars in Cincinnati, and cheetahs in Miami. Playback sessions were presented to these four pairs with both names being spoken in the name condition. Coders were instructed to code the behavior of each easily recognizable individual separately. They were told the position of each subject at the start of the session so that they could track the target individual. The subjects were actively monitored for the duration of each session for stress behaviors, such as repeated aggressive reactions, pacing, or other signs of distress. No abnormal behaviors indicating stress due to the playback sessions were observed in any of the subjects.

In the pilot study, each cat participated in six 3-trial playback sessions, with each session consisting of one playback from each speaker (MF, LF, UF). Whether playbacks included the cats' name was randomly determined for each session. Each speaker's voice was presented three times in each name condition across the course of the study. Each three-trial session lasted for 28 min total; three minutes of baseline prior to each playback, and three minutes of observation following the end of each playback with a minimum of five minutes between each trial (*e.g.*, 3 min observation–Playback 1–3 min observation +5 min ITI–3 min observation–Playback 2–3 min observation +5 min ITI–3 min observation–Playback 3–3 min observation). If more than one session was conducted for the same subject in a day, each session occurred at least one hour following the previous session, and no more than three sessions were conducted in a single day.

We conducted the main study approximately 3–6 months following the pilot study (depending upon the institution). In the main study, we used the habituation-dishabituation paradigm (*Saito & Shinozuka, 2013*). The playbacks for each session were in the order of unfamiliar voice 1, unfamiliar voice 2, unfamiliar voice 3, most familiar

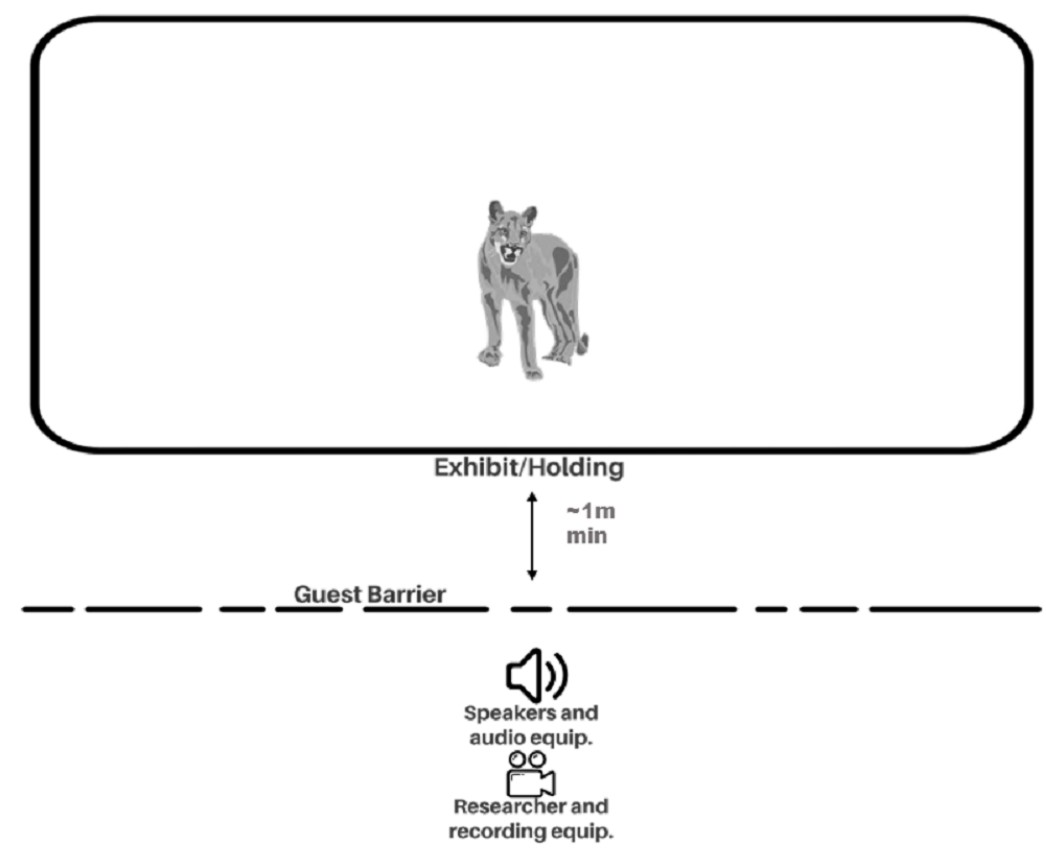

**Figure 1** Experimental set-up for playback trials and observation.

voice, unfamiliar voice 4 with 30 s of interstimulus silence between each voice. Unfamiliar voices were randomized in order between NA and NP sessions. Each cat participated in two playback sessions (one NA and one NP with order randomized across subjects). Sessions lasted no more than 3.5 min.

Behaviors were coded from video recordings of each session by two naïve coders using Boris software (*Friard & Gamba, 2016*). The behaviors coded were those inferred to indicate interest or attention to the location of the playback, such as change in gaze direction (a change in the direction of the eyes without movement of the head), ear and head movement (notable movement of the ears, either one or both, and the head that could not be accounted for by other behaviors such as grooming), locomotion towards or away from the sound (intentional locomotion that was initiated following the playback), or response vocalizations (social vocalizations such as chuffs, hisses, growls, or chirps, immediately following the playback). Behaviors were recorded for three minutes before the playback, during the playback and for three minutes following the end of each playback. Pre-trial recording allowed the establishment of baseline behaviors for the cat prior to the introduction of the cue. Recording for an extended period after the playback also gave us the ability to code for latency (how long after the recording did the reaction take place), intensity (how strong of a reaction was there), and follow up behaviors. A lack of response

was recorded as a latency of 30 s, the amount of time between each playback, and a duration of 0 s.

Intensity was rated on a scale of 0–4 with 0 indicating no reaction, and 4 indicating a full head turn and ear movement towards the speaker or locomotion towards the speaker. Intermediate scores were assigned if there was a mild head movement or a single ear twitch. Both positive and negative response behaviors were assessed, as per the Standardized Ethogram for Felidae (*Stanton, Sullivan & Fazio, 2015*). A negative behavior was defined as an aversive reaction to the sound, and included movement away from the sound, aggressive movement towards the sound (charging or bluff-charging), or aggressive vocalizations (hisses, roars, growls), although the only negative behaviors recorded were species-typical hissing from two serval subjects. To aid the accuracy of vocalization coding, coders were provided with auditory examples of observed vocalizations, as well as their context. The auditory examples were separate from the video recordings coded. The coders were trained on a test video to identify possible behaviors that may be observed in each video, as well as to train them on the BORIS software. Both coders were naive to any difference in the voice playbacks, only being informed that there were three or five playbacks that would occur within a session (depending on the study), and what words the playback would include for accurate identification.

# RESULTS

All analyses were conducted in IBM Statistical Package for the Social Sciences (SPSS) v. 28.

## Pilot study
### Reliability

To assess the reliability of our behavioral coding, each video was coded by two coders. Results were compared between a random selection of 25% of the sessions. We found excellent agreement between the two coders for latency (Pearson's $r = .998, p \leq .001$), duration ($r = .997, p \leq .001$), and intensity ($r = .900, p \leq .001$).

### Response to playbacks

Data were examined and met the requirements for sphericity, skewness and kurtosis. A repeated-measures ANOVA of latency to respond with name (absent, present) and familiarity (unfamiliar, less familiar, most familiar) along with their interaction as within-subjects factors revealed a significant interaction between familiarity and name ($F_{2,10} = 6.082$, $p = .019$, $\eta_p^2 = .549$). To explore the interaction, we conducted separate ANOVAs for name present and absent conditions. There was no significant main effect of familiarity when the cats' names were spoken, ($F_{2,10} = 3.544$, $p = .069$, $\eta_p^2 = .451$) but the effect was significant when the names were not spoken ($F_{2,10} = 4.747$, $p = .030$, $\eta_p^2 = .442$). Cats responded most quickly to the most familiar voice in both name conditions; However, the pattern of results differed slightly (see Fig. 2). If the name was spoken, the cats responded more quickly to both familiar voices compared to the unfamiliar voice, but if the name was not spoken, cats responded more quickly to the most familiar and unfamiliar voices compared to the less familiar voice. If we conducted the same analysis omitting

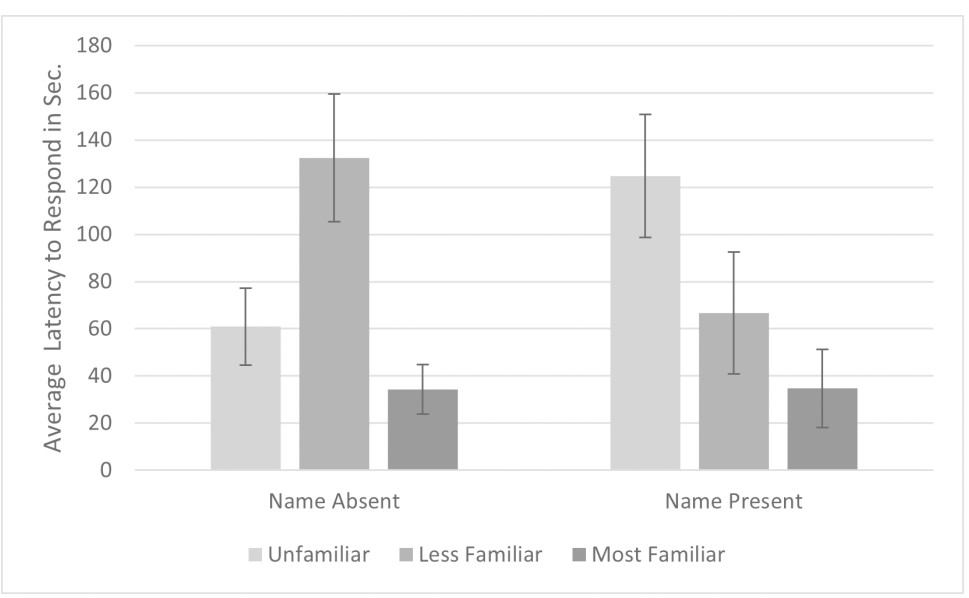

**Figure 2** **Average latency to respond to playbacks in the pilot study.** Note. Error bars indicate standard deviations. Latency to respond was the time in seconds from when the playback began to when the cats responded by behaviors such as ear movement and head turns. Seven cats were presented with voices that varied in familiarity (unfamiliar, less familiar, most familiar) and either spoke the subjects' name (Name Present) or did not (Name Absent).

the least familiar condition, we obtained a significant effect of familiarity ($F_{1,5} = 7.895$, $p = .038$, $\eta_p^2 = .423$), with no significant interaction with or main effect of name. Cats responded significantly more quickly to familiar ($M = 34.487s$, $SE = 13.191s$) *versus* unfamiliar voices ($M = 92.808s$, $SE = 10.479s$).

A repeated-measures ANOVA of intensity of response with name (absent, present) and familiarity (unfamiliar, less familiar, most familiar) along with their interaction as within-subjects factors revealed no significant effects. If we conducted the same analysis omitting the least familiar condition, we obtained a significant effect of familiarity ($F_{1,5} = 6.958$, $p = .046$, $\eta_p^2 = .582$). Cats responded with greater intensity to the familiar ($M = 1.527s$, $SE = .111s$) *versus* the unfamiliar voices ($M = 1.124s$, $SE = .109s$).

A repeated-measures ANOVA of duration of response with name (absent, present) and familiarity (unfamiliar, less familiar, most familiar) along with their interaction as within-subjects factors revealed no significant effects regardless of whether we included the least familiar condition in the analysis.

## Main study
### Reliability

To assess the reliability of our behavioral coding, each video was coded by two naïve coders. Results were compared between a random selection of 30% of the sessions (half name present and half name absent). We obtained an excellent level of agreement between the two coders for latency (Pearson's $r = .968, p \leq .001$), duration ($r = .913, p \leq .001$), and intensity ($r = .978, p \leq .001$).

### Response to playbacks

Data were analyzed using a mixed-model ANOVA for each of the outcomes of latency, intensity, and duration of responses. The within-subject variables of name use (NA, NP) and familiarity (playback trials 1–5), as well as the between-subject variables of rearing history (mother-reared, hand-reared), sex, and subfamily were examined. We included all two-way interactions involving the within-subjects variables of name and familiarity. We could not examine species differences due to the small sample sizes within each species. Sphericity and homogeneity assumptions were met for all outcomes. However, the data were not normally distributed according to Shapiro–Wilks tests and examination of Q-Q plots. We elected not to transform the data given that the data were not skewed and did not demonstrate kurtosis, and given the concern that transformation can obscure the interpretation of the results. Lastly, non-normality typically does not alter the validity of the results for F tests when homogeneity conditions are met (*Stevens, 2016*). The complete results appear in Table 2.

For latency, a main effect was found for subfamily ($F_{1,20} = 9.112$, $p = .007$, $\eta_p^2 = .790$), with subjects from Felinae responding more quickly ($M = 7.505s$, $SD = 1.369s$) than subjects from Pantherinae ($M = 13.360s$, $SD = 1.767s$). A significant effect was also found for familiarity ($F_{4,80} = 3.691$, $p = .008$, $\eta_p^2 = .156$) and name ($F_{1,20} = 4.571$, $p = .045$, $\eta_p^2 = .186$). Simple contrasts indicated that the cats responded significantly more quickly to the fourth playback trial (the familiar voice) compared to each of the other trials (see Fig. 3). Cats responded significantly faster when the name was absent ($M = 8.465s$, $SE = 1.543s$) *versus* present ($M = 11.357s$, $SE = 1.688s$). Omitting the data from the three lion subjects, there were still significant main effects of familiarity ($F_{4,68} = 4.788$, $p = .002$, $\eta_p^2 = .220$), name ($F_{1,17} = 5.109$, $p = .037$, $\eta_p^2 = .231$) and subfamily ($F_{1,17} = 13.314$, $p = .002$, $\eta_p^2 = .439$) in the same directions.

For intensity, a significant effect was found for familiarity ($F_{4,80} = 10.542$, $p \leq .001$, $\eta_p^2 = .345$). Simple contrasts indicated that the cats responded with significantly more intensity to the fourth playback trial (the familiar voice) compared to each of the other trials (Fig. 4). There was also a significant main effect of subfamily ($F_{1,20} = 8.185$, $p = .010$, $\eta_p^2 = .290$) with subjects from Felinae responding more intensely ($M = 2.131$, $SD = .204$) than subjects from Pantherinae ($M = 1.308$, $SD = .215$). There were no other significant effects or interactions for intensity. Omitting the data from the three lion subjects, there were still significant main effects of familiarity ($F_{4,68} = 9.273$, $p < .001$, $\eta_p^2 = .353$) and subfamily ($F_{1,17} = 16.802$, $p < .001$, $\eta_p^2 = .497$) in the same directions.

For duration, a significant effect was found for familiarity ($F_{4,80} = 4.021$, $p = .005$, $\eta_p^2 = .167$). Simple contrasts indicated that the cats responded for significantly longer to the fourth playback trial (the familiar voice) compared to each of the other trials (Fig. 5). There were no other significant effects or interactions for duration. Omitting the data from the three lion subjects, there was still a significant main effect of familiarity ($F_{4,68} = 3.902$, $p = .007$, $\eta_p^2 = .187$). However, there was also a significant interaction of name by subfamily, ($F_{1,17} = 5.852$, $p = .027$, $\eta_p^2 = .256$). To examine this interaction, the analysis was re-run for each subfamily separately without including subfamily as a factor. The main
**Table 2 Results from the mixed ANOVAs of latency, intensity and duration of response for the main study.**

| | Latency | | | Intensity | | | Duration | | |
|---|---|---|---|---|---|---|---|---|---|
| | $F$ | $p$ | $\eta_p^2$ | $F$ | $p$ | $\eta_p^2$ | $F$ | $p$ | $\eta_p^2$ |
| Subfamily | 9.112 | .007 | .313 | 8.185 | .010 | .290 | 0.241 | .629 | .012 |
| Sex | 1.319 | .264 | .062 | 3.621 | .072 | .153 | 0.367 | .552 | .018 |
| Rearing | 1.032 | .322 | .049 | 1.532 | .232 | .071 | 0.129 | .129 | .111 |
| Name | 4.571 | .045 | .186 | 0.119 | .734 | .006 | 0.147 | .706 | .007 |
| Familiarity | 3.691 | .008 | .156 | 10.542 | <.001 | .345 | 4.021 | .005 | .167 |
| Name by Familiarity | 0.318 | .865 | .016 | 1.990 | .104 | .090 | 0.370 | .829 | .018 |
| Rearing by Name | 2.223 | .152 | .100 | 0.659 | .426 | .032 | 2.219 | .152 | .100 |
| Rearing by Familiarity | 1.192 | .321 | .056 | 1.494 | .212 | .070 | 0.409 | .801 | .020 |
| Sex by Name | 0.856 | .366 | .041 | 0.595 | .449 | .029 | .000 | .998 | .000 |
| Sex by Familiarity | 1.578 | .188 | .073 | 1.358 | .256 | .064 | 0.249 | .910 | 0.12 |
| Subfamily by Name | 0.02 | .883 | .001 | 0.764 | .392 | .037 | 3.067 | .095 | .133 |
| Subfamily by Familiarity | 1.661 | .167 | .077 | 1.026 | .399 | .049 | 0.228 | .922 | .011 |

**Notes.**

$\eta_p^2$ denotes partial eta squared.

effect for name was not significant for either subfamily so this result will not be discussed further.

## DISCUSSION

Across two studies, various species of non-domesticated cats showed evidence of differentiating familiar human voices from unfamiliar voices, similar to what has been shown in domestic cats (*Saito & Shinozuka, 2013*) and hand-reared cheetahs (*Leroux et al., 2018*). This is the first time such an ability has been demonstrated in nine additional exotic cat species (Table 1), and the first time that early socialization and sex have been examined as possible contributors to responsiveness to human voices in exotic cats. Overall, cats responded more quickly and with greater intensity to familiar voices compared to unfamiliar voices regardless of sex, rearing and whether their names were spoken. In the main study, cats also responded for significantly longer to the familiar *versus* the unfamiliar voices. Evidence of the ability to differentiate familiar and unfamiliar human voices was not driven by the inclusion of the single gregarious species of cat as the significant effects of familiarity remained when we omitted data from our three lion subjects. Whereas lions may be the only truly social species of cat, all cat species must interact with other animals regularly; whether it be while hunting prey, raising young, finding a mate, or competing for territory. No cat lives a completely asocial life so social behaviors may still be beneficial. *Elbroch et al. (2017)* showed that even relatively asocial pumas (*Puma concolor*) were impacted by changes in the composition of their nearest neighbors. Their social interactions were explained better by reciprocity than by kinship, suggesting an ability to track social behaviors not often attributed to cat species (*Vonk, 2018*). The present results add to the growing literature suggesting that researchers have misattributed a lack of social cognition to non-group-living species and highlights the need to extend studies of social cognition to less commonly studied species.

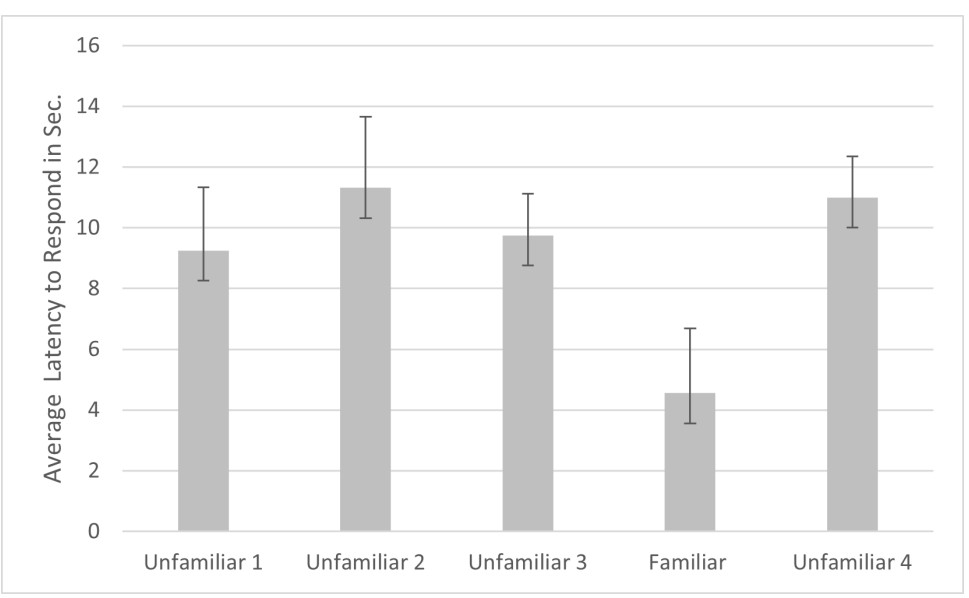

**Figure 3 Average latency to respond to playbacks in the main study.** Note. Error bars indicate standard deviations. Latency to respond was the time in seconds from when the playback began to when the cats responded by behaviors such as ear movement and head turns. In each session, 24 cats were presented with four different unfamiliar voices on trials 1–3 and 5 and a familiar voice on trial 4.

Research with domesticated species had suggested that the ability to recognize individual humans may be a consequence of domestication (*Adachi, Kuwahata & Fujita, 2007*; *Bensoussan et al., 2019*; *d'Ingeo et al., 2019*; *Saito & Shinozuka, 2013*), whereas studies with wild, yet highly social species suggest that this ability stems from the selective pressures associated with living in social groups (*Kriengwatana, Escudero & Ten Cate, 2015*). Exotic cats are neither domesticated nor highly social (other than lions). Of the 25 cats tested, only three were lions. Most were housed individually, and more than a third were raised by their mothers. Results from the only other study of this kind with exotic cats (cheetahs, *Leroux et al., 2018*) suggested that early socialization may play a role in the cats' abilities to discriminate human voices. The results from our main study, which are consistent with those of *Saito & Shinozuka (2013)* using the same habituation-dishabituation paradigm that those authors used with domestic cats, suggest a family-wide ability that is not dependent on domestication or social living. The lack of significant effects of rearing history suggests that this ability to discriminate human voices may depend upon regular rather than early exposure to humans. It is important to note that all but two of our subjects were reared in captivity (not wild born; the exceptions are two cougars at the Cincinnati Zoo) and that most of the hand-reared cats had transitioned to protected contact, a management strategy in which keepers can interact with animals only through a barrier, early in life. However, six of the hand-reared cats remained in close human contact serving as ambassador cats to promote education at the zoos where they were housed. Therefore, it is possible that effects are enhanced by continued close contact with humans.

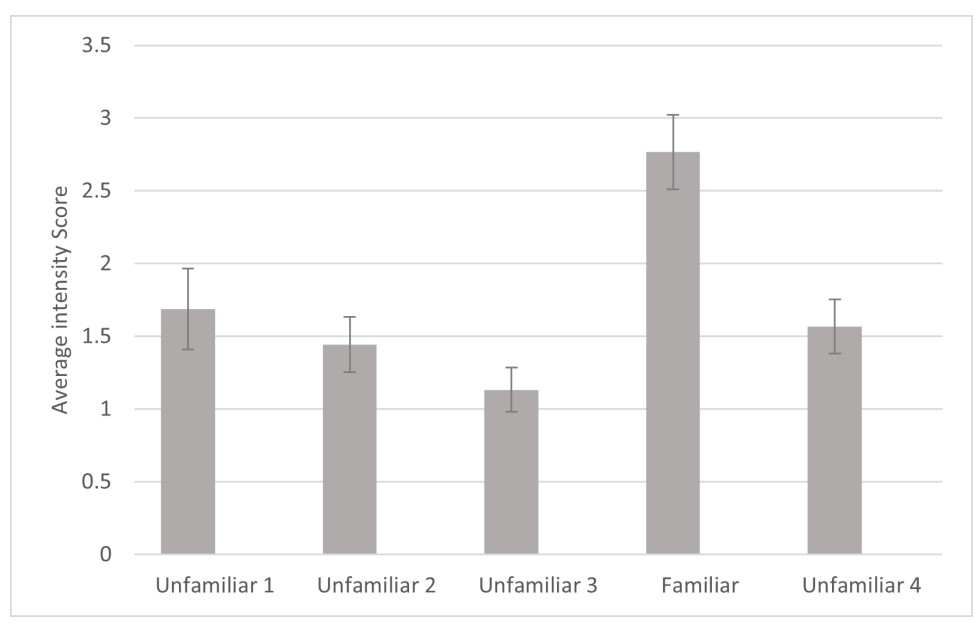

**Figure 4** **Average intensity of reaction as a function of playback trial in the main study.** Error bars indicate standard deviations. Intensity was measured from 0 (no response) to 4 (full head turn or locomotion towards speaker) of the first reaction following each playback as coded by human coders. In each session, 24 cats were presented with four different unfamiliar voices on trials 1–3 and 5 and a familiar voice on trial 4.

This is the first time the effects of the use of undomesticated cats' names, as spoken by familiar and unfamiliar voices, has been examined. Contrary to findings that domestic cats respond to their names (*Saito et al., 2019*), the use of the cats' names here had only a single effect on latency, and in the opposite direction to what we predicted with cats responding significantly more quickly when their name was not used. This finding that cats are not unduly distracted by the use of their names by strangers may aid zoological facilities in their decisions about whether to publicly post animal names. With one of the primary concerns being a degradation of the name cue in training if it is repeated frequently by guests without reward, the findings of this study suggest that cats may not be highly responsive to their names, but will be responsive to speakers with whom they are familiar. With studies showing that knowing the name of an animal, a form of anthropomorphism, makes people more sympathetic to the animal (*Chartrand, Fitzsimons & Fitzsimons, 2008*), and more willing to engage in conservation efforts (*Manfredo et al., 2020*), it may be beneficial for facilities housing exotic cats to post their names for visitors in an effort to engage them further with the conservation mission of the facility.

Most importantly, cats demonstrated differential responding to the familiar voice regardless of whether their name was spoken. Notably, all speakers spoke a familiar phrase but the cats responded with greater speed, more intensity, and for longer durations only when this familiar phrase was spoken by familiar voices. Although it is possible that they responded to specific phrasing rather than recognition of the speaker *per se* (*Kriengwatana, Escudero & Ten Cate, 2015*), the results nonetheless suggest that the cats recognize familiar

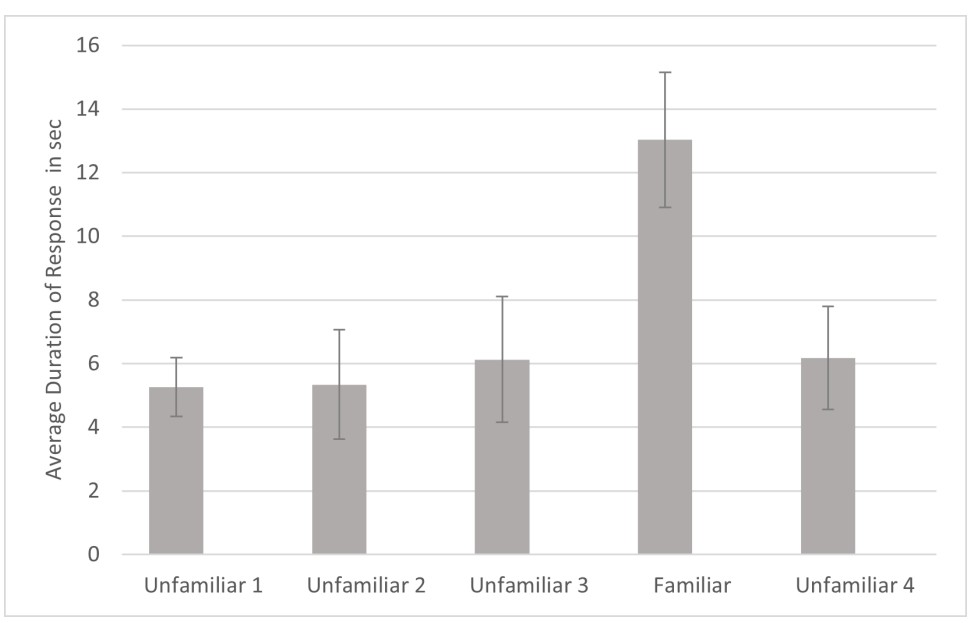

**Figure 5 Average duration of behavior as a function of playback trial.** Note. Error bars indicate standard deviations. Duration of response was the time from when behaviors such as ear movement and head turns began until they ended following a playback. In each session, 24 cats were presented with four different unfamiliar voices on trials 1–3 and 5 and a familiar voice on trial 4.

voices speaking familiar phrases. Future studies will need to determine whether cats also respond to familiar voices speaking unfamiliar phrases.

## LIMITATIONS AND FUTURE DIRECTIONS

Most of our subjects were housed individually. However, eight cats were tested in pairs so it is possible that some of these cats responded to their cage mate's response rather than to the playback itself. Although this study included representation from many cat species, it was limited by a small sample size within each species due to the relatively small population of captive exotic cats available for testing, precluding an analysis of species differences. Members of Felinae responded more quickly and with greater intensity compared to members of Pantherinae but subfamily did not interact with familiarity. We have no immediate explanation for the main effects of subfamily. Future studies will need to explore whether Felinae are generally more reactive than Pantherinae, which are typically larger. Future studies might examine whether time spent with cats, longevity of the human-cat relationship, and quality of the training interactions predict responsiveness to familiar *versus* unfamiliar caregivers. Importantly, the current results do not allow for the conclusion that cats can discriminate among individual humans. They merely show that cats respond more strongly to the voices of familiar *versus* unfamiliar humans in general. Future research is necessary to determine whether cats can discriminate between familiar and unfamiliar human scents and visual features. Methods such as cross-modal matching in which subjects are presented with stimuli from different modalities (*e.g.*, auditory, olfactory,

and visual stimuli) representing the same or different individuals and are expected to attend for longer to the mismatched stimuli compared to the matched stimuli (*e.g.*, *Takagi et al., 2019*) might demonstrate more conclusively whether cats discriminate among individual humans.

## CONCLUSIONS

This study contributes to the growing literature suggesting that adapting to a social lifestyle and human domestication are not the only important factors in predicting social cognitive abilities even when considering the ability to read human communicative cues specifically. Exposure to humans may promote the development of abilities that researchers would not be able to observe in the wild, such as the ability of cats to discriminate familiar human voices. This study adds to the growing body of work showing that even non-domestic cats are not indifferent to familiar humans and may help dispel the notion that cats are aloof.

## ACKNOWLEDGEMENTS

This research would not have been possible without immense support from Kim Ellis and The Creature Conservancy, Zoo Miami, Erin Curry, Amy Thompson, and the Africa and Night Hunters Teams with the Cincinnati Zoo, and Lindsey Zarecky, Megan Hankins, and Carolyn Mikulskis with the Greensboro Science Center. We would also like to thank Cameron Ferguson, Danielle Scott, and Hunter Cahoon, who coded the videos.

### Funding
The authors received no funding for this work.

### Competing Interests
Jennifer Vonk is an Academic Editor for PeerJ.

### Author Contributions
- Taylor Crews conceived and designed the experiments, performed the experiments, analyzed the data, prepared figures and/or tables, authored or reviewed drafts of the article, and approved the final draft.
- Jennifer Vonk conceived and designed the experiments, analyzed the data, prepared figures and/or tables, authored or reviewed drafts of the article, and approved the final draft.
- Molly McGuire performed the experiments, authored or reviewed drafts of the article, supervised data collection and coding, and approved the final draft.

### Animal Ethics
The following information was supplied relating to ethical approvals (i.e., approving body and any reference numbers):

The experiments reported here were reviewed and approved by Oakland University's IACUC

## Data Availability

The data for Study 1 and Study 2 (latency, intensity and duration) and codebook for Study 1 are available in the Supplemental Files.

All data are also available at OSF: Vonk, Jennifer. 2024. "Catcalls." OSF. January 21. doi: http://dx.doi.org/10.17605/OSF.IO/9VUK5.

## Supplemental Information

Supplemental information for this article can be found online at http://dx.doi.org/10.7717/peerj.16904#supplemental-information.

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
