# Peer review of "Catcalls: exotic cats discriminate the voices of familiar caregivers"

_PeerJ, doi:10.7717/peerj.16904_

## Round 0.1 · original submission · Major Revisions

Thank you for submitting this interesting study to PeerJ. I regret that I am unable to accept the manuscript for publication, at least in its present form. However, I am prepared to consider a new version that carefully takes into account the concerns and limitations highlighted by the reviewers. These need to be addressed in detail in a new version. Such a revised manuscript is likely to be reviewed again and there is no guarantee of acceptance. When you revise the study, please prepare a detailed explanation about how you have dealt with all the reviewer comments, as well as my own ones.

Overall, the standard of writing needs to be improved. For example, the introduction, lines 50-73, covers different topics. Reviewer 2 also provides detailed constructive advice.
See further advice here: The art of writing science
https://onlinelibrary.wiley.com/doi/full/10.1002/pro.514
"The first sentence of each paragraph should tell the reader what you expect them to get out of the paragraph that follows, which makes their job of following it far easier."
"A paragraph should discuss only a single idea"

Please check the whole text and remove dashes (e.g. L468, 484). Instead, write full, concise sentences. Currently, some of the text is more akin to email text.
L133. Formatting errors in references.
L499. The text would read much better if the science, the cats, their behaviours or their cognition, were the subjects of the sentences, and not the references. Please check the whole text and revise where possible.

**Language Note:** The Academic Editor has identified that the English language must be improved. PeerJ can provide language editing services - please contact us at copyediting@peerj.com for pricing (be sure to provide your manuscript number and title). Alternatively, you should make your own arrangements to improve the language quality and provide details in your response letter. – PeerJ Staff

Reviewer 1 ·

Basic reporting

The language is fine and I have no concerns.

The literature is well cited and with sufficient context.

I would suggest including further tables especially in the Results where there is no full report of the ANOVA models.

The introduction is clear, well-presented, and includes sufficient detail from suitable references. It does a good job of presenting the central research question and providing background to it. I would suggest you make it more explicit which species you are discussing however, as it takes until the methods in line 226 to list them. Considering the degree of interest that at least three of your species attract, it would be beneficial to make them a more prominent part of both the abstract and introduction. At the moment, this is buried.

Experimental design

Thank you for the opportunity to review this interesting paper. It addresses a current hole in the literature on vocal recognition – that of non-social species responding to heterospecifics. Unfortunately, it has one key flaw that limits its impact. The voices used all presented a phrase that was familiar to the animals, thus it may not be that they discriminated a familiar voice as related to a person so much as responded to the phrase as a sound with positive associations for that phrasing. It’s a shame that the choice was between the same phrase with or without the name presented, rather than a phrase that was unknown to the animals. While I am strongly of the opinion that the cats did discriminate the familiar voices, this study does not and cannot show that as they only discriminated between the presentation of a familiar and less-familiar version of the same familiar stimulus, e.g., the same phrase produced by familiar or unfamiliar humans.

Methods – I really like much of your design and controls. However, throughout, as noted above, it’s a shame that the control phrase was “name absent” rather than “meaningless phrase” as this limits the finding. If this could be added as a third condition for at least some of the cats, it would considerably strengthen the paper. This would involve re-testing a number of the same animals or testing new ones with a control phrase of something that the animals had never heard (potentially in a foreign language to ensure unfamiliarity). Without this, vocal discrimination is not shown.
In study 1, I’m slightly confused by the approach in the methods using “less familiar” which is still familiar as this seems likely to occlude the results due to the small sample size, and the lack of clarity as to whether the cats would care about degree of familiarity. It would be worthwhile exploring the binary of not familiar vs familiar for this as well as in study 2.
Also, while I understand the temptation, you overstate the significance of your p-values repeatedly. p = 0.052 is not a significant result at p < 0.05. Effect sizes are a better choice here as they do not rely on a somewhat arbitrary line for significance.

Study 2 – This is well designed in terms of protocol, but I’m confused by the statement that “All audio files were edited so that the playbacks all played with 1 dB and 2 Hz of the familiar voice using Audacity audio editing software.” Did you modify the frequency of the unfamiliar voices? This seems like a strange approach to me. The protocol is otherwise well-written, clear, and well-designed other than the phrase issue. Though it would be good to know when the name vs non-name phrases were presented and in what order.
The results here need to be presented in a clearer way with all results, not just significant ones, presented and a nice table summarising the effects from the mixed ANOVAs. The results themselves are interesting but the current presentation doesn’t help the reader to easily tease out some of the more complex relationships. I think that name didn’t matter, but it’s difficult to tell.

In the discussion, which is generally excellent, the sociality of the cats is discussed but this isn’t clearly modelled in study 2. If you remove the lions from the analysis, do the results change significantly? This would be an interesting additional analysis to include.

Validity of the findings

I have no concerns about the validity of the findings in so far as they can be supported but there are some issues with the conclusions drawn being slightly extended compared to the data due to the lack of a meaningless control phrase in testing. The statistics seem appropriate and properly performed.

While I applaud (loudly!) the sentiment in this conclusion, I’m not sure that you do show that they form relationships here. It would be better to conclude that the cats are not indifferent to us or aloof from us. It would be better to summarise that you have found strong evidence that cats respond more to the familiar stimulus and thus they may be forming social associations with humans in ways that should make us reflect again on our perception of their social cognition.

I would like to finish by saying this paper has been a pleasure to read and review, with a really interesting central thesis. I believe with a little more data added as described above (control condition with non-meaningful phrase used and the animals retested on their responses), it can be made considerably more robust and of wider interest. As it stands, it is still of interest but much more limited.

Additional comments

Keywords – a minor point here but I would suggest looking at your keywords again as they are not currently optimised to represent the paper.
52-57 combines both vocal recognition (e.g., kitten cries) and call recognition (e.g., dolphin signature whistles). I’d be a little more careful here in delineating the two as they are not the same. Please revise accordingly.
136 – The Latin name formatting has gone slightly wrong here.
154 – Hines is listed as n.d. – formatting error?
295 – Just to say congratulations, that’s a very high level of agreement between your two coders! It may be worth noting in the text that usually >0.8 is considered “excellent” to highlight this to your readers.
318 – I would be more cautious in wording here - 0.085 is not really approached significance and it’s better to report on the effect size here, so discuss the F not the p.
Figure 1 – I really like this. Well done.
390 – Can you please add how many sessions per enclosure total? I imagine some of the animals may have experienced habituation if they underwent several or if enclosures were near enough to have playbacks overheard.
Out of interest, why did you choose an ANOVA over a mixed effects model?
Again, congratulations on the high level of agreement.
435 – I would suggest slightly rewording this to discuss familiarity rather than playback trial.
It would really, really help the reader to have a full table here of the ANOVA results so that we can see at a glance what the significant effects were and how they differ across groups.
502 - Just to note I fully agree with your statement here on sociality and social cognition being under-researched.
560 – While I applaud (loudly!) the sentiment in this conclusion, I’m not sure that you do show that they form relationships here. It would be better to conclude that the cats are not indifferent to us or aloof from us. It would be better to summarise that you have found strong evidence that cats respond more to the familiar stimulus and thus they may be forming social associations with humans in ways that should make us reflect again on our perception of their social cognition.
Supplementary data sets – I’m confused by the labels given in CatcallsStudyOne csv. These don’t seem to match the code book. However, thank you for including the Codebook for study 2. This makes things clear and I am happy with the presentation of the data. Is there a reason that Study 1 has one csv file and study 2 has three?

·

Basic reporting

This study investigates the capacity of felid non-domestic species to discriminate between familiar and unfamiliar human voices. Using a habituation dishabituation paradigm, the authors demonstrate that felids display stronger reactions to familiar human voices. This study is an important contribution to the field and has the potential to provide further insights into the cognitive abilities of felids and shed light into the factors responsible for the emergence of such capacity. This could, in turn, help adapt husbandry and conservation policies. I believe however that the paper can be substantially improved before publication.

The most work to be done is to reduce the length of the paper, be concise, remove unnecessary sections. For instance, in the introduction, the paragraph from L88 to L110 could be remove without much impact on the paper nor much rephrasing necessary, that’s a clue that indicates this section does not add to the paper (although I absolutely love the last sentence of this paragraph). There are also a number of repetitions that should be avoided, e.g. the Leroux study mentioned on L82 but detailed on L124, or the methods from study 2 that repeats verbatim the methods described in study 1. In the latter case, the authors should probably revisit the structure of the paper, have a single methods section with sub sections dedicated to each study when the methods differ. In the same direction, I think the paper would benefit from restructuring the discussion: having only one discussion section, that is also more concise and straight to the point: the first part summarising the results of both studies, and then on the implication of the findings.

In the process of keeping the introduction concise, the authors should also clarify what is the bigger picture for the paper, what are the implications. For example, they mention in the intro that the study is important (L156) and will be beneficial (L163) without really specifying why it would be so. The authors try to get at this L165-170, but I find the justifications too general, under specific, without a clear message. Perhaps the authors should move away from the “applied” dimension of their research and stick to its fundamental aspect (this is only a suggestions, but developing and specifying the applied aspect could be equally interesting): this study helps better understand the factors influencing the emergence of such cognitive capacity. One could (but maybe shouldn’t?) even go extra and mention the social complexity hypothesis (Freeberg et al., 2012), with here a direct influence of early social experience on cognitive abilities. They come at this in the discussion more, which is good, but could be made more explicit in the intro.

There are also a number of less important – but still instrumental – points that merit attention:
- I would suggest restructuring the paper with study one framed as a pilot study. It is an interesting study, but 7 individuals for 5 species, this is a very small sample size (and I command the authors for acknowledging this L327), and although this is only part of the paper, it is confusing for the reader (and maybe more for a reviewer) to see such study taking that much space in the paper. We are worried for quite some time that study one is considered a core part of the paper, and therefore worried about the quality of the paper we are reading, while it all becomes clear when coming to study two. This is not to say that Study one is not interesting (also, it is a lot of work that has value), but framing it as a pilot study is more transparent and will lower the reader’s expectation for this specific section, which will help get the overall message of the paper across. Also maybe shortening the space dedicated to it – which will be easy if there is a unique methods section.
- For the statistical analyses, the authors use ANOVAs, which could make sense, but I am worried that given the small sample sizes, the distribution of the data might not be normal. The authors should provide a test verifying that (e.g. Shapiro). If the data is not normally distributed, the authors could do GLMMs using more or less the same structure as for the ANOVA. They would have to check their models afterwards (e.g. normality of residuals etc…), using for ex. the DHARMa package in R (function check.residuals).
- I am not super clear as to what is the added value of the “name present/absent”. It seems it is only investigated in Study one. I do not find anywhere an analysis on the effect of name in study 2, if I am not mistaken. I think this part might not be central to the core findings of the paper and I suggest to re-center the paper on familiar vs unfamiliar, and leave the effect of name for another study with more data to work with as well. This will help clarify the paper as well.

Finally, I provide here a bunch of minor corrections:
L52: Felis cactus
L65: This ability
L66-67: with researchers…cross-species communication, this sentence might benefit from rephrasing.
L71: be more important: I would be more careful and say it might be as important, if not more.
L113: As you note later (L136), cheetahs are included in the big cats from time to time. I would probably drop the “big cat” thing and stick to the morpho difference between Pantherinae and Felinae: i.e. the hyoid apparatus.
L133: problem with reference.
L144: higher production or more fear-related behaviours produced in…
L154: problem with reference.
L163: known to the public.
L210-212: the measures are somehow superficially described, it would help having more details, in the methods for example.
L264-265: The speakers were be controlled…
L286-287: we need more details and justifications.
L312: However, the pattern of results differed slightly; this phrase cannot be used as a standalone sentence.
L401: recorded as a latency.
L467: protected contacted?


I hope these comments and suggestions will help the authors improve the paper. I realise this might seem like a lot of work, but I would like to reiterate that I believe this paper represents already a substantial amount of work, that it is an important contribution to the field and that, once the authors restructured it so it is more concise and “straight to the point”, I am convinced it will become an influential piece.

Experimental design

The experimental design is neat and clear, adequate to the questions investigated. I simply would like to maybe have a bit more details into the behaviours measured (e.g. what was considered a change in gaze direction? A head turn? of minimum 15 degrees? 30? Etc) maybe alongside some literature justifying the choice of the authors. As mentioned above, the stats need additional work as well.

Validity of the findings

The findings are novel and will be influential in the field of animal cognition, sociobiology and evolutionary biology. As I stated above, I believe the discussions and conclusions could be more impactful if the manuscript is shortened, more concise.

---

## Round 0.2 · Major Revisions

Thank you for revising the submitted manuscript. The revised version was seen by one of the original reviewers, and one new reviewer (Reviewer 3). As you can see from the comments, there are still substantial changes that need to be made in order to improve the quality of the manuscript.

L123. Correct the formatting of the Reference.

Taking the Discussion as an example, at 6 pages, it is far too long, and some of the writing is still more thesis-like than paper-like standard. There are no references in the first three paragraphs of the Discussion. The purpose of the discussion is to discuss your results in light of previous research, and not simply to restate the results. It would be best to start with a discussion of major findings, and move the discussion of the pilot study to later. Revise phrases like "fairly small" and use more exact wording.

"The results from the main study were consistent with those of the pilot study". Revise text like this and make it more easy to understand by restating exactly what you mean.

L420. Use of "Thus" at the start of the paragraph. Please have a look at this very useful advice on scientific writing.
"The art of writing science: https://doi.org/10.1002/pro.514"; -
e.g. "Conceptually, a paragraph should also stand on its own two feet. For example, the first sentence of a paragraph generally should not refer to an idea in the paragraph above without, at the least, restating that idea." For this reason, avoid starting paragraphs with link words or phrases.

L435. "this ability". What ability?

Limitations and Future Directions. Please reduce to once concise paragraph.

"cats are aloof. As any zookeeper or cat owner will tell you, cats know their people, they just show it in their own way". Please revise and/or remove this text and try to take a more strictly scientific approach with the text rather than anecdotes.

As reviewer 3 has also pointed out, all the Tables and Figures need further information in order to make them more understandable.

·

Basic reporting

The authors responded extensively to my previous comments and suggestions and substantially modified their MS accordingly. I think the MS is much improved as a result.
I have only two additional points:
- First, the authors include a new analysis following the suggestion of R1, where they exclude lion data. This is not introduced neither in the introduction nor in the methods, this needs to be added.
- Second, although the authors did tremendous work to restructure the paper, I think it is still a bit lengthy. I suggest the authors further shorten the introduction and discussion to be even more concise and straight to the point.

Experimental design

The authors answered all my concerns and I appreciate the further details on the measurements taken and the stats implemented. I still think GLMMs would be better suited (it is possible to have nested designs with GLMMs as well), but I am convinced by the authors response.

Validity of the findings

The authors answered all my concerns.

·

Basic reporting

The language is largely clear and unambiguous, but there are some areas which can do with improvement (discussed in more detail in the pdf).

A good background for the current research is provided in the introduction section. There is at least one instance where arguments made by previous research are discussed without supplying a supporting reference and details of this can be found in the pdf provided.

The article structure is mostly professional, but could do with some improvement, for example, it is more conventional to discuss data analysis in a subsection of the methods, rather than in the results section (explained in pdf). The figures and tables can however use more significant improvement. For example, not enough information is given in the legends to interpret the figure, i.e., the units of measurement (i.e., seconds etc.) are not always shown, whether the figure is showing results from the pilot or main study is not always explicitly stated and whether the figure is showing the mean and standard error or some other type of descriptive measure is not explained. Raw data has been shared.

The results are relevant to the hypothesises laid out in the introduction section.

Experimental design

The background and research question has been well defined in the introduction section and the authors have identified a knowledge gap and explained how their research fills it.

There are obvious constraints in performing experiments on zoo animals, but I would consider the author has performed experiments to a rigorous enough technical and ethical standard around these constraints.

Methods are largely described in sufficient detail, but there are key bits of information missing. These have been addressed in more detail in the pdf provided, but the authors have not for example, mentioned the software they performed the analyses in, and although they briefly discuss it in the introduction section, the authors do not cover in any detail the specific predictions of the habituation-dishabituation paradigm used in the main study either in the methods section or later on.

Validity of the findings

Data has been provided and the analysis although unconventional (many authors would have used mixed models for such an experiment), is likely valid (although I do have a few questions to be addressed in the pdf).

I consider the authors interpretation of their results to be in-depth and relevant, although the discussion section could be a bit more concise.

Additional comments

Please refer to the attached pdf.

---

## Round 0.3 · Major Revisions

Thank you for carrying out many of the required changes but unfortunately more needs to be done to address previous concerns. Please revise all Figure legends or Table headings that are not detailed enough. Figures and Tables complement the text. Their content must be so precise that they can be viewed and understood without reading the text of the manuscript.
e.g.
Table 2. Results from the main study
Figure 2. Latency to Respond to Playbacks in Study One
Figure 2, 3 & 5. Y-axis label is not appropriate.
The previous letter states: "AU We have edited tables, figures and their captions as recommended." This has not been done as stated, and therefore please check very carefully, once again, all Tables, Figures, Axes, and related texts. Please look at some published papers in PEERJ as that might help guide you.

Note, Keywords should normally be in alphabetical order.

Reviewer 1 ·

Basic reporting

I'm a bit confused to why they are presenting bar charts rather than boxplots but other than that, this is all fine.

Experimental design

No comment. I'm satisfied.

Validity of the findings

No comment.

Additional comments

I'd like to thank the authors for their careful attention to my comments in round 1. The changes they have made are excellent and a lot of work has gone into improving the manuscript.

The language is clear and well-written throughout.

Incredibly minor change at line 275 - could you please just remind the reader the coding is in Boris? You state this above but in line 199.

I really like the new analysis that looks at the data without lions and would like to congratulate the authors on adding this in such a robust way.

The paper is valuable, well-written, clearly presented, well-designed, and highly interesting. Well done!

---

## Round 0.4 · accepted · Accept

"We do not understand why you say the y axis labels are not appropriate for Figures 2, 3 and 5 because they report exactly what was graphed (e.g., “Mean Latency to Respond in Sec”)." "in sec" is not an appropriate, scientific way to denote scientific units on a Figure. But if you feel strongly about using that format, go ahead.